# Development of an Integrated Salt Cartridge-Reverse Electrodialysis (Red) Device to Increase Electrolyte Concentrations to Biomedical Devices

**DOI:** 10.3390/membranes12100990

**Published:** 2022-10-13

**Authors:** Efecan Pakkaner, Jessica L. Orton, Caroline G. Campbell, Jamie A. Hestekin, Christa N. Hestekin

**Affiliations:** 1Ralph E. Martin Department of Chemical Engineering, University of Arkansas, 800 W. Dickson St., BELL 3202, Fayetteville, AR 72701, USA; 2Department of Biomedical Engineering, University of Arkansas, John A. White, Jr. Engineering Hall, 790 W. Dickson St. Suite 120, Fayetteville, AR 72701, USA

**Keywords:** reverse electrodialysis, biopower cells, salt cartridge, ion-exchange membranes, salinity gradient, blue energy, portable devices

## Abstract

Emerging technologies in nanotechnology and biomedical engineering have led to an increase in the use of implantable biomedical devices. These devices are currently battery powered which often means they must be surgically replaced during a patient’s lifetime. Therefore, there is an important need for a power source that could provide continuous, stable power over a prolonged time. Reverse electrodialysis (RED) based biopower cells have been previously used to generate continuous power from physiologically relevant fluids; however, the low salinity gradient that exists within the body limited the performance of the biopower cell. In this study, a miniaturized RED biopower cell design coupled with a salt cartridge was evaluated for boosting the salt concentration gradient supplied to RED in situ. For the salt cartridge, polysulfone (PSf) hollow fibers were prepared in-house and saturated with NaCl solutions to deliver salt and thereby enhance the concentration gradient. The effect of operational parameters including solution flow rate and cartridge salt concentration on salt transport performance was evaluated. The results demonstrated that the use of the salt cartridge was able to increase the salt concentration of the RED inlet stream by 74% which in turn generated a 3-fold increase in the open circuit voltage (*OCV*) of the biopower cell. This innovative adaptation of the membrane-based approach into portable power generation could help open new pathways in various biomedical applications.

## 1. Introduction

Implantable biomedical devices are increasingly common due to advancements in nanotechnology and biomedical sciences. Biomedical implantable devices, including cardiac pacemakers, artificial organs, drug pumps, and sensors, require continuous stable, and reliable power to operate, creating a demand for a safe, reliable, and stable power source. The power consumption of these devices is relatively low. For example, a cardiac pacemaker requires a power input ranging between 8–400 μW that is provided by a Lithium-ion battery [1]. The average life span of these batteries ranges from six to nine years [2], and therefore often require surgical replacement during the lifetime of the patient. [2] This surgery comes with significant risks for the patient including infections and bleeding [3,4]. In addition, battery replacement surgeries put a heavy burden on the medical system. For example, pacemaker replacement surgeries currently occupy more than 25% of all implantation surgeries performed nationally [5]. Thus, an alternative method of powering up these medical devices is highly desirable.

Several efforts have been previously made to develop alternative power sources to increase the longevity of biomedical device operations. One approach harnessed solar energy to be potentially converted into electrical power [5]. Although sufficient energy could be harvested, a major drawback was the incompatibility of solar panels with the human body. Another study explored harvesting energy from the everyday movements of the human body by kinetic and thermal energy harvesters [1]. Although the human body produces enough kinetic energy to be harnessed for powering devices in the lower energy range, the power output was unstable including unpredictable and sudden interruptions, making the kinetic harvesters unreliable power sources for implantable biomedical devices [6,7]. Several biologically based approaches, including enzymatic and microbial fuel cells, have also been explored as promising power production techniques [8]. Although numerous studies have shown that they could produce enough power to be independent power sources [9,10,11], the major challenge is the short lifetimes due to enzyme instability and degradation. These disadvantages in all the previous approaches indicate that there is still a need for a viable long-term solution for replacement batteries.

Reverse electrodialysis (RED) is a salinity-gradient energy-capturing technique where the potential energy of mixing is harnessed through a series of ion-exchange membranes and electrodes [12]. It is a promising, clean way of harvesting the Gibbs Free Energy of Mixing by combining two streams containing different salinities to produce usable electric energy [13]. Other major advantages, including higher overall efficiency and applicability for hyper-saline (>40 g/kg) conditions, make RED advantageous when compared with the other salinity-gradient energy methods (e.g., capacitive mixing and pressure retarded osmosis). This is especially true when natural water sources are used, including seawater and brackish water [14]. Efforts have already been made to produce clean energy using RED on a large scale in the Netherlands where a power plant uses natural water sources to produce up to 1 MW [15]. Another advantage of RED is that it can harness natural water sources for power production using non-invasive measures [12]. A previous study showed that a net power density of 1.2 W/m^2^ was achieved by only exploiting seawater and river water as the RED feed streams [16]. Although the development of RED technology is mainly based on an increased number of stacks to maximize the power density, miniaturized RED power sources have also shown promising results in the areas of application including biosensors [17], transdermal drug delivery [18], and MEMS (microelectromechanical systems)-based power generation [19]. Moreover, a recent study by Pakkaner et al. used biologically relevant fluids as a source to produce power under physiological conditions [20]. The same study demonstrated that the natural ion gradient between the renal artery and vein, maintained through the complex process of removing waste from the body as urine, could be used for energy production via a miniaturized RED power cell. The disadvantage of the proposed approach as a power source was the comparatively small concentration gradient found naturally in the kidney led to low power density (PD). However, it was observed that enlarging the salinity gradient between the two physiological streams led to a significant increase in power output. Therefore, for RED to be a feasible method of power generation for biomedical devices, it is necessary to develop an approach to enhance the salt concentration of one of the feed streams going into the RED power cell.

In this study, a salt pick-up cartridge integrated into a RED power cell device was designed and tested. It aimed to boost the natural salt concentration gradients found in the body through an in-situ salt cartridge. The designed salt cartridge would then be coupled with a RED power cell to generate a large enough power density for biomedical devices. This study explores the parameters affecting salt uptake from the salt cartridge.

## 2. Materials and Methods

### 2.1. Materials

Sodium chloride (NaCl, ACS reagent ≥ 99%) and N-Methyl-2-pyrrolidone (NMP, ACS grade) were purchased from VWR USA (Radnor, PA, USA). Polysulfone resins (~60 kDa) were acquired from ACROS Organics (Antwerp, Belgium). Titanium wires (0.01 inch diameter) and platinum electrodes (gauze, 100 mesh 99.9%) were bought from Alfa Aesar (Haverhill, MA, USA). Cation exchange membranes (Fumasep FKL-PK-130 CEM, 130 μm) and anion exchange membranes (Fumasep FAA-PK-130 AEM, 130 μm) were acquired from Fumatech BWT GmbH (Bietigheim-Bissingen, Germany). All solutions were prepared using Milli-Q water (18 MΩ cm).

### 2.2. Hollow-Fiber Membrane Production and Salt Cartridge Preparation

Polysulfone (PSf) hollow-fiber membranes that were used in the cartridges were made in-house as previously described [21]. PSf casting solution was prepared by adding solid PSf resins into N-methyl pyrrolidone (NMP) to reach the final concentration of 18% by weight. The casting solution was kept roller-mixed for 4–5 days until no solid clumps were present in the solution. The solution was then vacuum-filtered through ceramic filters to remove any impurities and air bubbles. Hollow fibers were cast via non-solvent phase-inversion spinning through a single-orifice spinneret. Polymeric casting and bore solutions (15% NMP in water) were introduced into the spinneret with pressurized gas sources (2–3 psig). While the polymer solution creates the outer layer of the fibers, the bore solution flows through the internal capillary to promote the hollow structure. The solution flow rates and pressure were then optimized to achieve the correct fiber consistency that is then collected on the spinneret inside the fiber water bath. The fibers formed were then collected onto a spinning roll, then soaked, and stored in 70% ethanol solution until use.

The salt cartridge was designed to permit electrolytes from a concentrated donor salt (NaCl) solution within the fibers to diffuse into a feed acceptor stream to increase its salinity. This increased salinity feed acceptor solution next flows from the cartridge and into the RED biopower cell as the concentrate stream (Figure 1). In order to prepare the salt cartridge unit, cast hollow fibers were rinsed with de-ionized water and placed in a clear PVC pipe, and potted with epoxy glue at the ends of the pipe (Figure 1c). SEM pictures taken of the hollow fibers revealed that the membranes exhibited a porous structure with an average fiber diameter of 150 μm (Figure 1b). The unit was dried for 24 h to allow the adhesive to be solidified and sealed with PVC caps. Hollow fibers were checked for leaks by flowing N_2_ through them underwater. The cartridge unit was excessively rinsed with de-ionized water, followed by conditioning with concentrated NaCl solutions. Finally, salt cartridges were loaded and conditioned with high-concentration donor solution overnight, and fibers were equilibrated. The conductivity values of the donor and acceptor solutions were measured before and after the experiments, and the salt concentrations were calculated accordingly. The aqueous content of both donor and acceptor solutions were marked before and after the experiments, to confirm the absence of any leaks or water transport between two phases.

### 2.3. Salt-Pick-Up Experiments

The in-situ salt diffusion performance of the cartridge was tested using several device and operational parameters. Firstly, the device integrity and pre-compression tests were conducted by using aqueous NaCl and de-ionized water solutions as donor and acceptor sides, respectively. Pressurized NaCl solution was pumped through the hollow fibers (donor side) into the cartridge (acceptor side), where deionized water was circulated, and the conductivities of both solutions were measured before and after the experiment. At the end of the runs, no internal or external leaks were observed in the device.

For salt pick-up experiments, four different feed (7.2 g/L of NaCl) flow rates were tested, including 0 (no flow case), 1, 20, and 100 mL/min. The feed concentration (7.2 g/L NaCl) was selected because it mimics a typical salt concentration in the blood [20]. The feed solutions were pumped through the cartridge using MasterFlex peristaltic pumps and pump heads (MasterFlex 7014-21, Cole-Palmer, Vernon Hills, CT, USA). The effect of the hollow fiber surface area on mass transfer was also investigated using two types of salt cartridge units with 10 and 20 fibers, respectively. Finally, the influence of the donor salt concentration was investigated using varying salt concentrations (20, 100, and 350 g/L). Since the solubility limit of NaCl in water is around 360 g/L at room temperature, 350 g/L was selected as the highest salt concentration for the study to prevent precipitation [22]. Samples were collected after 6 h, and NaCl concentrations of the solutions were measured with inductively coupled plasma (ICP) (iCAP Q, Thermo Scientific, Waltham, MA, USA) and conductivity measurements (Fisherbrand, Thermo Scientific, Waltham, MA, USA). The *average salt transport* pick-up was calculated with the equation given below:(1)$$average\;salt\;transport\;\left(\%\right)=\frac{C_{final}-C_{initial}}{C_{initial}}*100$$
where; *C_final_* and *C_initial_* are final and initial NaCl concentrations of the acceptor solution, respectively.

### 2.4. Reverse Electrodialysis (RED) Setup and Salt-Cartridge Coupling

The RED biopower cell setup was assembled as previously described [20]. Briefly, a miniaturized 3-D printed RED stack was used as the biopower cell. The dilute stream (7.2 g/L NaCl) was split off from the feed into the overall unit (salt cartridge + RED biopower cell). The concentrate stream was split off from the feed into the overall unit and then introduced in situ by being pumped through the salt cartridge as the feed acceptor stream to create a higher salinity solution through the diffusion of salt from the donor salt solution contained within the hollow fibers. By splitting off part of the overall feed and flowing it through the salt cartridge a higher salinity gradient between the dilute and concentrate streams to the RED stack was created, which should result in an increased power density (Figure 1). Based on the parametric salt pick up studies, the donor solution concentration was chosen to be 350 g/L, and a 20-fiber cartridge design was used. The device was operated for 24 h continuously, with a solution flow rate of 1 mL/min. A physiological temperature (~37 °C) was maintained for all the flowing streams using two heating mantles controlled via PID controllers (Autonics, Mundelein, IL, USA). The RED power cell’s voltage was measured using a multimeter (Klein Tools MM-700, Lincolnshire, IL, USA).

## 3. Results and Discussion

### 3.1. Salt Pick-Up Studies

Reverse electrodialysis is a membrane-based electrochemical technique that exploits the salinity gradient between two streams to harness sustainable blue energy. The chemical potential captured is converted into electrical energy at the electrodes via redox reactions [23]. One approach to increase the power density of a RED device is to increase this salinity gradient. This study increased the salinity gradient in situ through the use of a salt cartridge unit.

A series of preliminary salt-pick experiments were performed to determine the effect of the salinity difference on salt diffusion from the donor solution to the acceptor solution. The chosen donor solution flow rate was 1 mL/min, for the cartridge that was equipped with 10 fibers. Salt was diffusively transported from the high salt (20 or 100 g/L NaCl) donor solution to the feed (acceptor) solution which was at the physiologically relevant salt concentration of 7.2 g/L. The average amount of salt transport was found to be around 14% for the low salt donor solution (20 g/L) while for the high salt donor solution (100 g/L) the average salt transport was almost quadrupled, with an average increase of 51% (Table 1). In both cases, the feed solution increased in salinity due to the salt cartridge as was desired.

The effect of the feed (acceptor) flow rate on the salt transport was investigated by testing four different flow rates (0, 1, 20, and 100 mL/min) as shown in Figure 2. The selection of the intermediate flow rate values (1 and 20 mL/min) was based on physiological values, as the human artery and vein average blood flow rates were previously measured to be 3–26 and 1–4.8 mL/min, respectively [24]. For the case of a stagnant feed solution (Figure 2a), the salt transport was very low and was not influenced by an increase the concentration of the donor salt solution. This is most likely due to diffusive limitations at the interface of the donor and acceptor solutions leading to a decrease in the local salt gradient over time. Specifically, this mass transfer limitation could be attributed to the un-disturbed stagnant layer on the membrane surface, which is less perturbed by the lower flow rates, making the effect of the diffusion distance more dominant, as well as with the increased boundary layer resistance that arises from the salinity gradient that occurs in between bulk and the membrane-liquid interface [25]. In addition to its low mass transfer, stagnant transport is also undesirable since it promotes bacterial growth in the form of biofilms [26]. For the cases with a continuously recycled feed solution (Figure 2b–d), the increase in the salinity of the acceptor solutions from 20 to 350 g/L showed a more than four-fold increase in salt transport for the higher flow rate cases of 20 and 100 mL/min (Figure 2c,d). At these higher flow rates, there is a smaller diffusive boundary layer near the membrane surface which leads to a higher salt transport.

In addition, the effect of the mass transport area (number of fibers) can be observed by comparing the red (10 fibers) vs. black (20 fibers) data in Figure 2. The number of fibers in the cartridges were chosen based on a variety of considerations including unit size, cost, and potential hold-up volume of physiological fluids. Although the mass transfer area is doubled from ~0.006 m^2^ to ~0.012 m^2^ when the number of fibers is doubled, the salt transport does not double. Interestingly, similar results have been seen for dialyzers, where increasing the membrane area does not have a linear effect on uremic toxin removal. A study by Maduell et al. revealed that increasing the dialyzer area from 1 m^2^ to 1.4 m^2^ and 1.8 m^2^ did not translate into urea removal at the same magnitude [27]. The change in the urea removal capacity was just 1% and 3% for 1.4 and 1.8 m^2^, respectively. Moreover, the authors also claimed that the main factors limiting the diffusion in their case were the blood flow rate and solute concentration. A similar trend can also be seen in the current study.

For biologically compatible processes, it is necessary to have a laminar flow. Therefore, the Reynolds number for each flow rate case was calculated. For the largest flow rate (100 mL/min), the Reynolds number was calculated as ~90, which is well within the laminar flow regime [28]. Moreover, the Reynolds number of the shell-side fluids was observed to be highly effective on mass transfer coefficients for hollow-fiber membrane cartridges, as a reported study by Satoru et al. revealed that solute mass transfer coefficients were found to be exponentially increased related to dialysate Reynolds number [29]. Figure 2a shows the stagnant case where the pick-up solution was loaded, and transport of the electrolytes was observed. Compared to the dynamic runs, the pick-up was an order of magnitude lower for all concentration cases, with 20-fiber units shown to have slightly higher transport rates. It has been previously shown that the increase in the blood flow rate inversely influences diffusive solute transport in the blood flow concentration [30]. However, the rate of salt transport was found to be increasing with a decreasing rate, which could be related to the equilibration of the acceptor solution by the cartridge.

### 3.2. Reverse Electrodialysis Operation with Salt Cartridge

Increasing the molar concentration of the concentrate feed stream to the RED is highly effective in increasing the power density of the stack, as previously published studies reported that a 10-fold increase from 0.5 to 5 M in the concentrate stream yields a 4-fold boost in the PD [31]. Thus, the RED biopower cell was operated with and without the integration of the salt-pick-up cartridge, and the corresponding open circuit voltage (*OCV*) values that were recorded around the power cell were plotted in Figure 3. As can be seen, almost a three-fold increase in the *OCV* was recorded in the presence of the cartridge after two hours of operation, while the *OCV* values showed an almost constant trend for the no cartridge operation. When the device was operated for 24 h continuously the change in the *OCV* was within the range of values observed at the 2 h point showing that it was stable both with and without the cartridge.

Taking the highest *OCV* values achieved in the study, the theoretical power output that would be expected from that RED unit was calculated via Ohm’s Law:(2)$$P=\frac{OCV^{2}}{R_{stack}}$$
where *P* is the theoretical maximum power, and *R_stack_* is the stack resistance for the given operating conditions which can be determined from the following equation [32]:(3)$$R_{stack}=\frac{N}{A}*\left(R_{AEM}+R_{CEM}+\frac{d_{C}}{ConcC}+\frac{d_{D}}{DiluteC}\right)+R_{electrodes}$$
where *N* is the number of cell pairs, *A* is the membrane area, *R_AEM_*, *R_CEM_* are the anion- and cation- exchange membrane resistances respectively, *d_C_ d_D_*, are dilute and concentrate channel thicknesses respectively and *ConcC* and *DiluteC* are the concentrate and dilute stream concentrations, respectively. The maximum theoretical power density that could be delivered by the power cell was calculated using the equation below:$$P_{d,max}=\frac{OCV^{2}}{4*A*R_{stack}}$$

The representative power demand of a commercial standard pacemaker was previously given in the study by Haeberlin et al. [33]. In the mentioned study, it was concluded that the average demand for a commercialized pacemaker was around 40 μW/cm^2^. For a membrane area of 27 mm^2^ and dilute and concentrate channel heights of 0.5 mm, the maximum power and power density that the RED power cell could deliver were found to be around 10 μW and 38.5 μW/cm^2^, respectively. Therefore, based on the power studies, the integrated salt cartridge RED device is capable of powering up a biomedical device solely on the salinity gradient energy, by exhibiting a power output performance in the right order of magnitude.

Theoretically, assuming that no adverse effects of membrane fouling are observed, for a 5-year continuous operation of a pacemaker with 8 μW of energy input demand, a total of 1262 Joules of energy should be implemented salt-cartridge integrated RED cell. Knowing the fact that 1 eV is equal to ~2 × 10^−19^ Joules [34], a simple calculation was performed to estimate how much salt was needed to support the pacemaker continuously. It was calculated that the cartridge added a total of 7.41 g NaCl salt to the RED concentrated solution. This amount was feasible when the physiological concentrations were taken into consideration, as the upper limit of a healthy blood Na^+^ level for humans was given as 145 mEq/L [35]. Also, considering an average human consists of 42 L of water, the total Na^+^ present in the body stream sums up to 140 g in total. Moreover, a study published in the American Society for Nutrition has reported that the upper limit for a Na intake that could lead to acute toxicity was 1 g of Na/kg body weight [36]. Therefore, it could be concluded that an addition of 7.41 g over 5 years can be facilitated without adverse side effects. Moreover, the integrated device could open a promising pathway for the devices in the lower-energy scale, including drug pumps and sensors, which could be tested in situ for the future remarks of this study.

## 4. Conclusions

A miniaturized salt cartridge was built to improve the performance of a miniaturized RED unit. The effect of different parameters on the salt transport performance in the salt cartridge unit was discussed and analyzed. Higher concentration gradients between the feed and donor solution within the cartridge unit led to a higher average salt transport in the cartridge. In addition, it was possible to alter the amount of salt transport by changing the flow rate of the feed stream coming into the cartridge. A 50% salt increase could be achieved with moderate NaCl concentrations for the donor solution such as 100 g/L. The salt cartridge unit was then connected to an RED power cell and demonstrated an increased production of power. In the future, studies will focus on the long-term capabilities of salt cartridge performance, influence on the RED power output, and 3D printing to minimize and integrate the salt cartridge design.

## Figures and Tables

**Figure 1 membranes-12-00990-f001:**
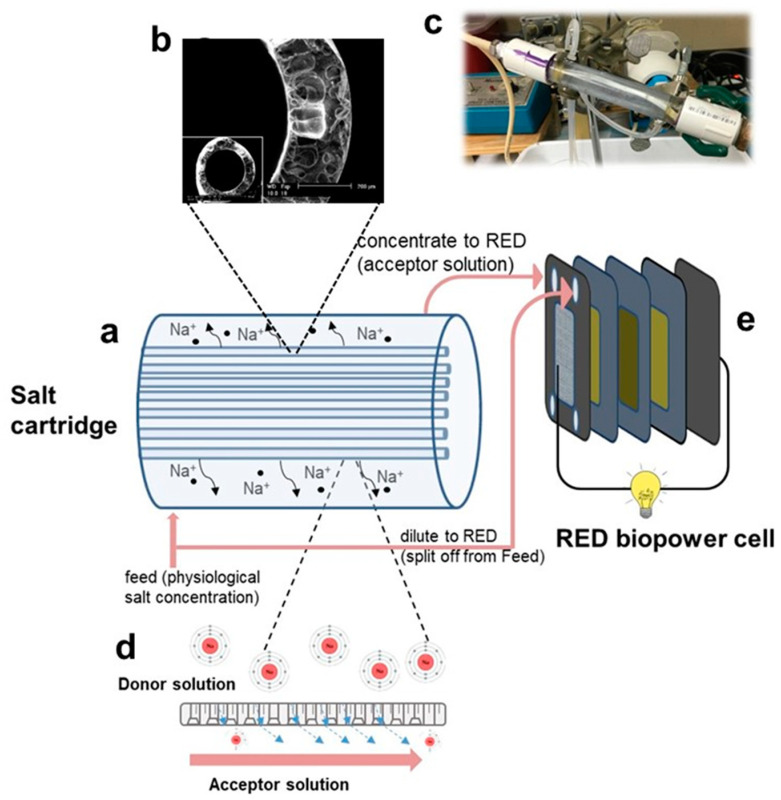
Schematic representation of the salt cartridge-RED power cell set up: (**a**) schematic drawing of the cartridge illustrating ion movement, (**b**) SEM image of the hollow-fiber membranes used in the cartridge, (**c**) picture of salt cartridge, (**d**) a schematic close up of the transport of Na^+^ ions through the hollow-fiber membranes, (**e**) schematic drawing of the RED biopower cell.

**Figure 2 membranes-12-00990-f002:**
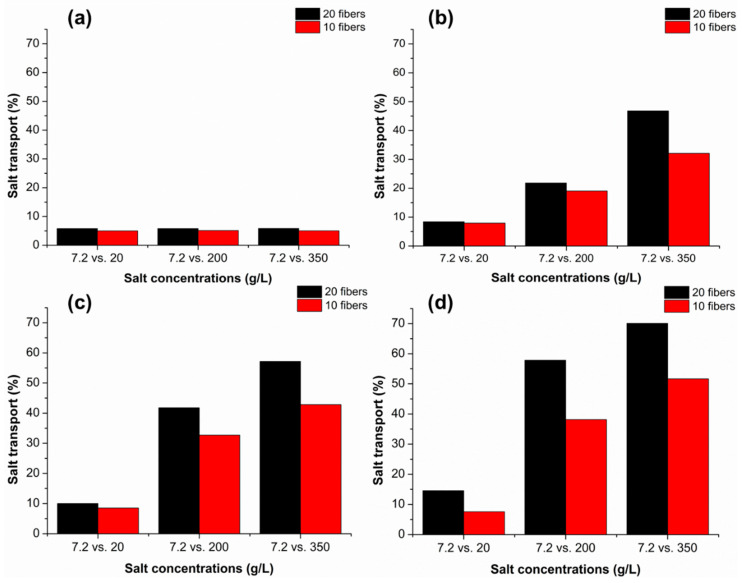
The effect of solution flow rate on the salt pick-up under different salinity gradients in the salt cartridge. Solution flow rates include: (**a**) 0 mL/min, (**b**) 1 mL/min, (**c**) 20 mL/min and (**d**) 100 mL/min.

**Figure 3 membranes-12-00990-f003:**
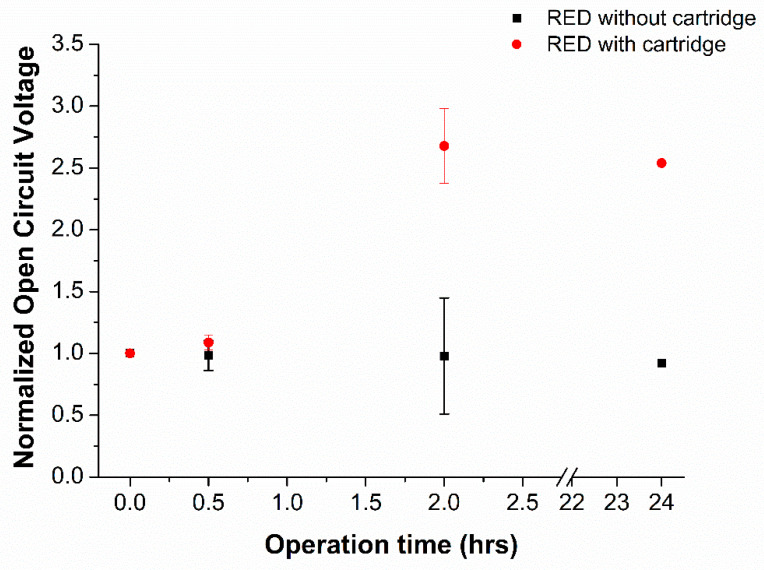
Open circuit voltage (*OCV*) values recorded for RED biopower cell operations with and without the salt cartridge. N = 3 for 0–2 h and n = 1 for 24 h.

**Table 1 membranes-12-00990-t001:** Preliminary pick-up tests for function verification * (n = 3).

Feed (Acceptor) NaCl Concentration (g/L)	Cartridge (Donor) NaCl Concentration (g/L)	% Average Salt Transport	Acceptor Final NaCl Concentration (g/L)
7.2	20	14.08 ± 7.3	9.37 ± 0.05
7.2	100	51.07 ± 24.3	29.59 ± 0.92

* Q_soln_: 1 mL/min, #fibers: 10, V_cartridge_: 30 mL.

## Data Availability

Data available on request due to privacy restrictions. The data presented in this study are available on request from the corresponding author.

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
