# Peer review of "Development of an Integrated Salt Cartridge-Reverse Electrodialysis (Red) Device to Increase Electrolyte Concentrations to Biomedical Devices"

_membranes, 2022, doi:10.3390/membranes12100990_

Round 1
Reviewer 1 Report
In this study, a miniaturized RED biopower cell design coupled with a salt cartridge was proposed to increase the open circuit voltage (OCV) of the RED system. The authors found that the introduction of a salt cartridge was capable of increasing the electrolyte concentrations, thus leading to an almost 3-fold increase in the OCV of the biopower cell. This manuscript is generally well organized and the conclusions are supported by the results. Some minor concerns should be clarified.
1. The results parts are scarcity in data. Only three figures were provided. If possible, please supply some data.
2. The authors need to clarify the differences between the directly adding the salts in the concentrate chamber of the RED stack and the usage of a salt cartridge. The direct addition of salt can also increase the concentration gradient and is much simpler and more convenient.
3. In Fig. 2, it is necessary to specify why only 10 fibers and 20 fibers were considered.
4. In Fig. 3, the OCV data of the other time intervals should be provided. It is necessary to specify the changing trend of the OCV after 2 hours.
5. Is the outputting OCV stable when the salt cartridge was supplied?
Author Response
Dear Reviewer 1,
You can find the comments & responses regarding your comments on our manuscript. We would like to thank you for your kind attention and contribution to the study.

Reviewer 2 Report
This manuscript is devoted to the study of a RED power cell device coupled with a salt cartridge, which is expected to be used in future as a possible energy source for biomedical devices, for example cardiac pacemakers. The authors conducted a study on the effect of several device and operational parameters, such as hollow fiber surface area, concentration of donor solution and flow rate of the feed stream coming into the cartridge, on salt transport in the salt cartridge unit. This study is fully consistent with the subject of the journal. It is obvious that the authors performed a large amount of experimental work, but the article looks rather modest. Only after some improvements, I can recommend this work for publication.
1) In the section 3.2. Reverse electrodialysis operation with salt cartridge, there is no description of the parameters of the salt cartridge (number of fibers, concentration of donor solution, flow rate of the feed stream), which was integrated into the RED power cell device for comparative analysis. According to the logic, salt-pick-up experiments were carried out in order to find the optimal parameters, which must be taken into account in conducting the main studies on reverse electrodialysis operation.
2) In my opinion, two hours of operation of the RED biopower cell is not enough in the context of this study. I believe that the data presented in Fig. 3 is necessary to supplement with the results of experiments with longer operation time to evaluate the effect of the integrated salt cartridge on the RED power output.
3) In the article, there is some confusion over the concentration of donor solution values in the salt cartridge unit. Section 2.3 states that experiments were performed for three different salt concentrations: 20, 100, and 350 g/L. At the same time, in a series of preliminary salt-pick experiments, two concentrations (20 and 100 g/L) were investigated, and in the main salt-pick experiment, three other concentrations (20, 200 and 350 g/L) were studied. Could you please explain why you took some concentrations in the preliminary experiments and others in the main experiment?
4) I advise you to present the diagrams in Fig. 2, making values on the scale that shows salt transport (%) uniform. In its current form, the significant impact of solution flow rates on salt transport is not clearly evident, it is difficult to assess the real difference between diagrams a, b, c, d.
There are also some inaccuracies and typos in the article, for example:
1) All the abbreviations used must explained, even if they are generally accepted (for example, “MEMS”).
2) Under Table 1, there are designations that are not deciphered in the text of the article (Qsoln and others).
3) In the fragment d of Fig. 2, the name of the axis differs from fragments a, b, c. Check this.
4) Line 204: check the phrase “while for the high salt donor solution (10 g/L)” for typo. Seems like it should be “while for the high salt donor solution (100 g/L)”.
Author Response
Dear Madam/Sir,
Attached are the authors' replies and edits, regarding your comments and feedback on the manuscript. We would like to thank you for your kind attention and contribution to the current work.
Kind regards

Reviewer 3 Report
The manuscript prepared by Pakkaner et al. illustrated the feasibility of coupling a salt cartridge with RED to increase the power efficiency of RED. The motivation of this work was demonstrated well, and the manuscript was written with clear language. However, this work is barely a scientific article, as it only demonstrated an observation without in-depth analysis and sufficient data. Also, the idea of this work was not innovative, because increasing the concentration gradient surely means more power output in RED. I have listed my comments below for the author to consider.
1. Line 138 – 139, this is a poor illustration in SEM image. First, I don’t see any evidence to indicate the nanoporous structure of the membrane. Second, the SEM image is in low-resolution to see any meaningful observations. Third, based on Figure 1B only without other evidence to conclude a nanoporous structure of the membrane, to me, is an issue of attitude. Please pay serious attention to your evidence and corresponding analysis.
2. In my understanding, the salt (supporting electrolyte, NaCl) are acting as a carrier to send redox species to the electrode surface, because redox species are generally low-conductive compounds (e.g., ferrocyanide). Therefore, increasing the salt concentration surely enhances the power output as more redox species are carried to the electrode surface. What do you think of this in your work?
3. Also, to be more comprehensive, I suggest the authors perform more electrochemical measurements to demonstrate the relationship between supporting electrolyte concentration and power output. Such as EIS or CV.
4. Line 219 – 222, this statement is wrong. In a stagnant area, there will be no effect of concentration polarization, CP only occurs when the permeate flows through the membrane. In CP, a larger flux results in more severe CP effects.
5. Line 225 – 226, “since no convection……through nanopores”, this conclusion is supported without evidence. Again, the evidence present in the study is weak. I suggest the authors provide more comprehensive data to support your key conclusion.
6. The illustration of the figures (all figures) from the authors is poorly structured. Mostly, the author only delivered the conclusion from the figure without in-depth analysis. For example, Line 222 – 224, “……increasing the salt gradient between the donor and acceptor had a significant effect on salt transport”, can you provide more description of your data? Honestly, I moved, back and forth, between script and figures to figure out what you are trying to say.
7. For the observation that increasing the surface area is not effective to increase salt transport rate, the authors simply used another study to explain. Citing other works to explain your data without independent analysis is not acceptable. I strongly suggest the authors perform a deeper analysis and cite other work to support your conclusion if any.
8. Please validate Equation 3.
9. Line 281 – 305, for the data (10 uW, 38.5 uW/cm2), how did you calculate? Have you provided the equation for power density? From line 290 to 305, the conclusion lacks basic justification. I barely see a research work that could deliver their major conclusion without sufficient evidence.
Author Response

(The authors gave the same response as above.)

Round 2
Reviewer 2 Report
The authors revised the article and took into account all the comments and recommendations. Now, the article can be accepted for publication.
P.S. Line 293: Correct the typo "cartridg".
Reviewer 3 Report
I thank the author's response to my comments and recommend acceptance.